# Poincaré sphere trajectory encoding metasurfaces based on generalized Malus' law

Zi-Lan Deng [1,5] ✉, Meng-Xia Hu[1,5], Shanfeng Qiu[2], Xianfeng Wu[2], Adam Overvig [3], Xiangping Li [1] ✉ & Andrea Alù [3,4] ✉

As a fundamental property of light, polarization serves as an excellent information encoding carrier, playing significant roles in many optical applications, including liquid crystal displays, polarization imaging, optical computation and encryption. However, conventional polarization information encoding schemes based on Malus' law usually consider 1D polarization projections on a linear basis, implying that their encoding flexibility is largely limited. Here, we propose a Poincaré sphere (PS) trajectory encoding approach with metasurfaces that leverages a generalized form of Malus' law governing universal 2D projections between arbitrary elliptical polarization pairs spanning the entire PS. Arbitrary polarization encodings are realized by engineering PS trajectories governed by either arbitrary analytic functions or aligned modulation grids of interest, leading to versatile polarization image transformation functionalities, including histogram stretching, thresholding and image encryption within non-orthogonal PS loci. Our work significantly expands the encoding dimensionality of polarization information, unveiling new opportunities for metasurfaces in polarization optics for both quantum and classical regimes.

As one of the fundamental properties of light, polarization plays a crucial role in various fields of science and technology, including quantum optics[1,2], nonlinear optics[3], liquid crystal (LC) displays[4], 3D glasses, and all-optical computing[5–11]. Conventional approaches for general polarization manipulation of the incoming light typically rely on bulky optical components based on linear birefringence and/or diattenuation materials, leading to separate amplitude and phase retardation control under a linear polarization basis. Considering the polarization as an important information indicator, the encoding of polarization information is of particular interest in optical recording, displays, storage, and encryption[12]. As there is no direct means to detect the polarization state, Malus' law (ML) is usually used to map polarization information to intensity that could be readily perceived by photoelectric detectors,

and therefore forms the cornerstone for polarization information encoding in both classical and quantum scenarios[13]. Nowadays, polarization encoding in conventional platforms, such as image displays based on liquid crystals[14] or micro-wire grating polarizer arrays, are based on the projection relation between linear polarizations, as the conventional ML refers to the linear polarization basis. Recent advances in metasurfaces have enabled several breakthroughs in polarization control, enabling simultaneous amplitude and phase retardation modulation beyond linear polarization basis over an ultrathin and agile platform[15,16], greatly expanding the scope of polarization manipulation and empowering intriguing applications such as metasurface vectorial holography[17–21], quantum entanglement[22,23], and polarization imaging [24–27]. Metasurfaces also

[1]Guangdong Provincial Key Laboratory of Optical Fiber Sensing and Communications, Institute of Photonics Technology, College of Physics & Optoelectronic Engineering, Jinan University, Guangzhou 510632, China. [2]Shphotonics LLC, Suzhou 215000, China. [3]Photonics Initiative, Advanced Science Research Center, City University of New York, New York, NY 10031, USA. [4]Physics Program, Graduate Center, City University of New York, New York, NY 10016, USA. [5]These authors contributed equally: Zi-Lan Deng, Meng-Xia Hu. ✉e-mail: zilandeng@jnu.edu.cn; xiangpingli@jnu.edu.cn; aalu@gc.cuny.edu

provide a powerful platform for polarization information encoding in a pixelated level by treating each unit cell as a local waveplate or polarizer. In this way, a variety of so called Malus metasurfaces[28–32] have been proposed to realize advanced polarization information encoding for ultra-high resolution display[28], multiple channel multiplexing[30] and computational imaging encryption[33]. Although the polarization manipulation capabilities of metasurfaces have been largely expanded in recent times, polarization-intensity mappings have still been restricted to the conventional ML framework that describes the projection behavior of linear polarizations in a one-dimensional (1D) space. Therefore, information encoding based on Malus metasurface platforms have so far been limited to fixed orthogonal channels[15,16] or linear polarization projections[32], since the 1D projection space is not sufficient to capture the overall polarization mapping flexibility, hindering more sophisticated encoding capabilities.

Here, we introduce a Poincaré sphere (PS) trajectory encoding scheme for versatile polarization information encoding by employing a generalized Malus' law (GML) spanning the entire PS. The GML describes a universal projection rule from one arbitrary elliptical polarization to another, both of which could be located at arbitrary locations of the solid PS. Its expression manifests an infinite degeneracy in polarization projection, providing new degrees of freedom to realize arbitrary modulation mappings and enable parallel information channels. The modulation mapping trajectory on the PS can be either engineered by an arbitrary analytic function or by aligned grids provided by dual information channels, significantly enhancing polarization manipulation functionalities in terms of flexibility and generality. The additional dimension in the mapping relation is an uncharted degree of freedom that can be used to freely encode polarization trajectory and information in terms of both information location and content, leading to unprecedented opportunities for optical analog image transformation functionalities, which include image histogram stretching, thresholding and information location encryption beyond orthogonal channels. The principle of GML-equipped metasurfaces deals with the polarization and intensity relationship in a dimensionality-enhanced space. It is not only applicable to the metasurface platform, but also to other polarization optics platforms, including conventional liquid crystal (LC) displays and birefringent 2D materials, enabling new classical and quantum information encoding methodologies applicable to optical recording and cryptography.

## Results

The conventional ML describes the typical cosine square projection between two linear polarizations with azimuth difference $\Delta\psi = \psi - \psi_0$, as shown in Fig. 1a. A PS representation of the ML (inset of Fig. 1a) shows the output intensity (color) of all possible projections of incident linear polarization onto a linear polarizer with allowed state $|\alpha_0^{Lin}\rangle$ and stopped state $|\alpha_0^{Lin,\perp}\rangle$, both on the equator of the PS. To build the projection relation of arbitrary polarization pairs covering the full PS, one can preset a PS polarizer with allowed state $|\alpha_0(2\psi_0, 2\chi_0)\rangle$ that

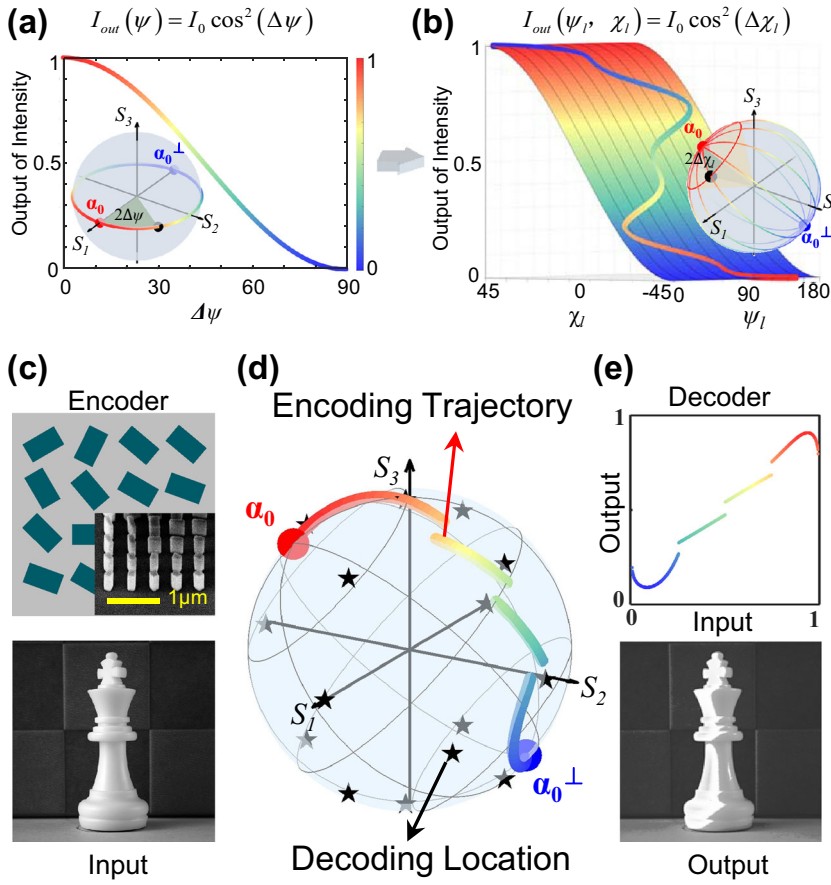

**Fig. 1 | Concept of PS trajectory encoding based on GML. a, b** Extension from ML (**a**) governing the projection rule with respect to azimuthal difference $\Delta\psi$ to the GML (**b**) governing the full PS projection rule with respect to local PS parameters $(\psi_l, \chi_l)$. The color represents the output intensity, the red/blue dots represent polarizer allowed/stopped state, and the black dot represents the incident polarization state (inset). **c–e** PS trajectory encoding based on GML: an arbitrary grayscale image is encoded into an (**c**) anisotropic metasurface with a (**d**) designed PS trajectory (color curves) between the local north/south poles (red/blue dots). Various image decoding styles are performed by adjusting analyzer state to different PS locations (black stars) covering the full PS, leading to engineered histogram transformation function and output image shown in (**e**).

could locate on arbitrary positions of the PS, where $\psi_0, \chi_0$ are the azimuth and ellipticity parameters of the polarization state. And then a local PS system could be defined with $|\alpha_0(2\psi_0, 2\chi_0)\rangle$ and its orthogonal state $|\alpha_0^\perp(2\psi + \pi_0, -2\chi_0)\rangle$ as the local north/south poles, which plays the same role of right-handed circular polarization (RCP) and left-handed circular polarization (LCP) states in the global PS system, respectively. In the local PS system, the output intensity by projecting the state $|\alpha_l\rangle$ onto the PS polarizer allowed state $|\alpha_0\rangle$ can be written as (Supplemental Note 1)

$$I_{out} = \left|\langle RCP|\alpha_l\rangle\right|^2 = A_0^2 \cos^2\left(\pi/4 - \chi_l\right). \tag{1}$$

This local GML expression follows the cosine square projection form in terms of the parameter $\Delta\chi = \pi/4 - \chi_l$, which is the local ellipticity difference between the polarizer allowed state and the incident state. Interestingly, this equation indicates that the intensity of output light is solely dependent on the local ellipticity $\chi_l$, while the other local azimuth parameter $\psi_l$ provides an uncharted new dimension for polarization encoding with arbitrary trajectory, as shown in Fig. 1b. Inset of Fig. 1b shows that the output intensity follows the cosine square projection along each local longitude line by connecting the poles $|\alpha_0\rangle$ and $|\alpha_0^\perp\rangle$ in the local PS, while they remain constant along each local latitude line (trajectory with fixed $\chi_l$ and varying $\psi_l$). This projection relation can be considered as the extension from the plane angle projection (inset of Fig. 1a) to a solid angle projection on the sphere (inset of Fig. 1b).

By successively rotating the local PS by the angle $2\Delta\chi = 2\chi_0 - \pi/2$ with respect to the $S_2$-axis and by the angle $2\Delta\psi = 2\psi_0$ with respect to the $S_3$-axis, we can explicitly obtain the local PS parameters $(\psi_l, \chi_l)$ in terms of global PS parameters $(\psi, \chi)$, as well as the polarization basis parameters $(\psi_0, \chi_0)$ (Supplementary Note 1), and then express the GML in terms of global PS parameters as

$$I_{out} = \frac{A_0^2}{2}\left[\cos 2\chi_0 \cos 2\chi \cos(2\psi - 2\psi_0) + \sin 2\chi_0 \sin 2\chi + 1\right]. \tag{2}$$

This global form of GML provides the output intensity of an arbitrary polarization state $(\psi, \chi)$ projected onto another polarization state $(\psi_0, \chi_0)$. Note that, Eqs. (1) and (2) describe a GML for fully polarized light and polarizer located on the surface of the PS. The most general case of GML on the solid PS[34] involving partial polarized light and partial polarizer can be further deduced in the same framework as follows (Supplementary Note 1.2),

$$I_{out} = A_0^2 \frac{1 - pp_0}{1 + p_0} + A_0^2 \frac{pp_0}{1 + p_0}\left[\cos 2\chi_0 \cos 2\chi \cos(2\psi - 2\psi_0) + \sin 2\chi_0 \sin 2\chi + 1\right], \tag{3}$$

where, $p$ and $p_0$ are the degree of polarization (DoP) of the incident light and the partial polarizer, respectively. As expected, Eq. (3) collapses to Eq. (2) for $p = p_0 = 1$, and it further collapses to the original ML for $\chi_0 = 0$ and $\chi = 0$, where both the local north/south poles and the modulation path are restricted to the PS equator. We note that, although the routine mathematic procedure with Mueller matrix and Stokes vector treatment could also handle the universal projection relation overall the entire PS, it could hardly give the insight of hidden dimension assisted information encoding with a clear physical picture. In addition, as the anisotropic metasurface design with spatially varying width, length and orientation angle is always associated with the complex-valued Jones matrix polarization parameters, rather than the power united Stokes parameters, the Jones calculus of polarization projection law provides more straightforward means for metasurface design.

By modulating the intensity profile of an image pixel by pixel with spatially varying polarizations based on ML, polarization encoding spans a variety of applications such as LC Displays, 3D glasses and information encryption. Conventional ML only considers 1D projection, in which both the encoding path and decoding locations are restricted on the equator of the PS. This leads to a straightforward and simple reverse deciphering process, largely limiting the breadth of image transformation functionalities.

By contrast, polarization encoding based on the dimension-upgraded GML can generate much more possibilities. As for a given image profile $I(x,y)$ in Fig. 1c, only the local ellipticity parameter could be determined by $\chi_l(x,y) = \pi/4 - \arccos\sqrt{I(x,y)}$ following Eq. 1. The other untapped parameter $\psi_l(x,y)$ can be arbitrarily assigned as a function of $\chi_l$: $\psi_l = f(\chi_l)$, whose form could be an analytic function, piece-wise function, or digital sampled matrix, manifesting an modulation trajectory on the PS (Fig. 1d) for added functionalities (Supplementary Note 2). The encoding local parameters $(\psi_l, \chi_l)$ are then transformed to global parameters $(\psi, \chi)$ as follows for physical carrier implementation (Supplementary Note 3),

$$\psi = \frac{1}{2}\arctan$$
$$\left[\frac{\cos 2\chi_l \cos 2\psi_l \sin 2\chi_0 \sin 2\psi_0 + \cos 2\chi_l \sin 2\psi_l \cos 2\psi_0 + \sin 2\chi_l \cos 2\chi_0 \sin 2\psi_0}{\cos 2\chi_l \cos 2\psi_l \sin 2\chi_0 \cos 2\psi_0 - \cos 2\chi_l \sin 2\psi_l \sin 2\psi_0 + \sin 2\chi_l \cos 2\chi_0 \cos 2\psi_0}\right], \tag{4}$$

$$\chi = \frac{1}{2}\arcsin\left[-\cos 2\chi_l \cos 2\psi_l \cos 2\chi_0 + \sin 2\chi_l \sin 2\chi_0\right], \tag{5}$$

After the encoding process, the decoding process can be performed by employing the global form of GML (Eq. 2) with the basis parameters $(\psi_0, \chi_0)$ simply replaced by the analyzer parameters $(\psi_A, \chi_A)$, leading to the customized histogram transformations of the image shown in Fig. 1e at the analyzer PS locations indicated by the stars on Fig. 1d. The detailed flowchart of the encoding and decoding procedure could refer to Supplementary Figs. 9–12. For the encoding procedure, each pixel of the modulated image is mapped on the local ellipticity parameter $\chi_l(x,y)$ based on the local form of GML, while the other polarization parameter could either be modulated by setting $\psi_l$ as an analytic function of $\chi_l$ (Supplementary Fig. 9), or constructing another local PS system in which the local ellipticity parameter $\chi_l'(x,y)$ is used to modulate another image (Supplementary Fig. 11). Then, the local PS parameters $(\psi_l, \chi_l)$ or $(\chi_l, \chi_l')$ are transformed to global PS parameters $(\psi, \chi)$ with the information on polarization basis parameters $(\psi_0, \chi_0)$. After this procedure, the global parameters $(\psi, \chi)$ are mapped onto the meta-atom parameters for experimental realization. For the decoding procedure, the metasurface is illuminated with circularly polarized incident light, a spatially varying profile of polarization states appears in the transmitted light beam. Placing a PS analyzer with allowed polarization state $(\psi_A, \chi_A)$ on the transmitted light beam, the decoded intensity can be obtained by the global form of GML. Varying the analyzer state on the PS, a variety of decoded intensity information is captured by a CCD camera (Supplementary Figs. 10 and 12).

Compared with conventional ML polarization encoding, the PS trajectory encoding based on GML is much more flexible, as (i) the polarization basis poles, (ii) the form of trajectory between the poles and (iii) the decoding analyzer positions are released from the equator to the full surface of the PS (Supplementary Fig. 2). The histogram transfer function is determined by the combination of the above three factors (Supplementary Figs. 3–6 and Videos 1–3), providing a large library of image transformation operations. Note that there are also approaches based on cascading retarders and linear polarizers to encode incoming polarization states for high signal-to-noise full Stokes parameter measurement[35,36], yet the bulky cascading approach lacks the capability of pixel-leveled polarization encoding in a dimension-upgraded 2D space, limiting its range and flexibility. We stress that this encoding approach is applicable for various platform

consisting of pixelated anisotropic elements with arbitrarily controllable phase retardation and orientation angle of the fast/axis axes, such as nematic LCs, anisotropic 2D materials, and metasurfaces (Supplementary Figs. 7 and 8).

As a first experimental demonstration, we designed and fabricated a metasurface operating with circular polarization basis, mediated by the trajectory function $\psi_l = 2\cos^2(\chi_l - \pi/4)$ (Fig. 2). To experimentally decode the information, a decoder analyzer was realized by cascading a quarter-waveplate and a linear polarizer after the metasurface sample in the experimental setup (Supplementary Fig. 13). To filter out an arbitrary polarization state with PS parameters $(\psi_A, \chi_A)$, the orientation angle of the quarter-waveplate and the linear polarizer should be $\theta_{\lambda/4} = \psi_A$, $\theta_P = \psi_A - \chi_A$ (Supplementary Note 4). When the analyzer state $|\alpha_0(2\psi_A, 2\chi_A)\rangle$ coincides with the modulation basis states $|\alpha_0\rangle$ (blue star) and $|\alpha_0^{\perp}\rangle$ (green star), the histogram transfer functions are simply $y = \pm x$ ($x$: input, $y$: output), leading to the original and grayscale-reversed versions of the predesigned image (Fig. 2c, d). Moving the PS location of the analyzer, the histogram transfer function changes, based on different modulation relations between the input

and output images (Supplementary Fig. 5). In particular, at analyzing state $\left(\psi_A = 135^O, \chi_A = 35^O\right)$ (Fig. 2a, red star), we obtain an S-shaped transformation curve (Fig. 2b, red curve), which enables image contrast enhancement as the histogram can be stretched from a relatively narrow range to nearly the entire [0, 1] range (Fig. 2e).

Another image thresholding function was realized as shown in Fig. 3. The metasurface is encoded with a two-segment PS trajectory (Fig. 3a). The unitary (Fig. 3b, blue line) and anti-unitary (Fig. 3b, green line) transformations with original and gray-scale inversed output images (Fig. 3c, d) are decoded at the local north/south poles (Fig. 3a, blue/green stars). At the analyzing location of $(\psi_A = 35^O, \chi_A = 0^O)$ (Fig. 3a, red star), a thresholding input-output curve could be realized (Fig. 3b, red curve), The transformed histogram of the output image shows two gathering sites at the two ends, which confirms the binary feature of the image (Fig. 3e). The quantitative analysis of the experimental input-output image reconstruction is presented in Supplementary Fig. 14. In addition, the thresholding value can be adjusted by engineering the trajectory with different segment length combinations at will (Supplementary Fig. 15).

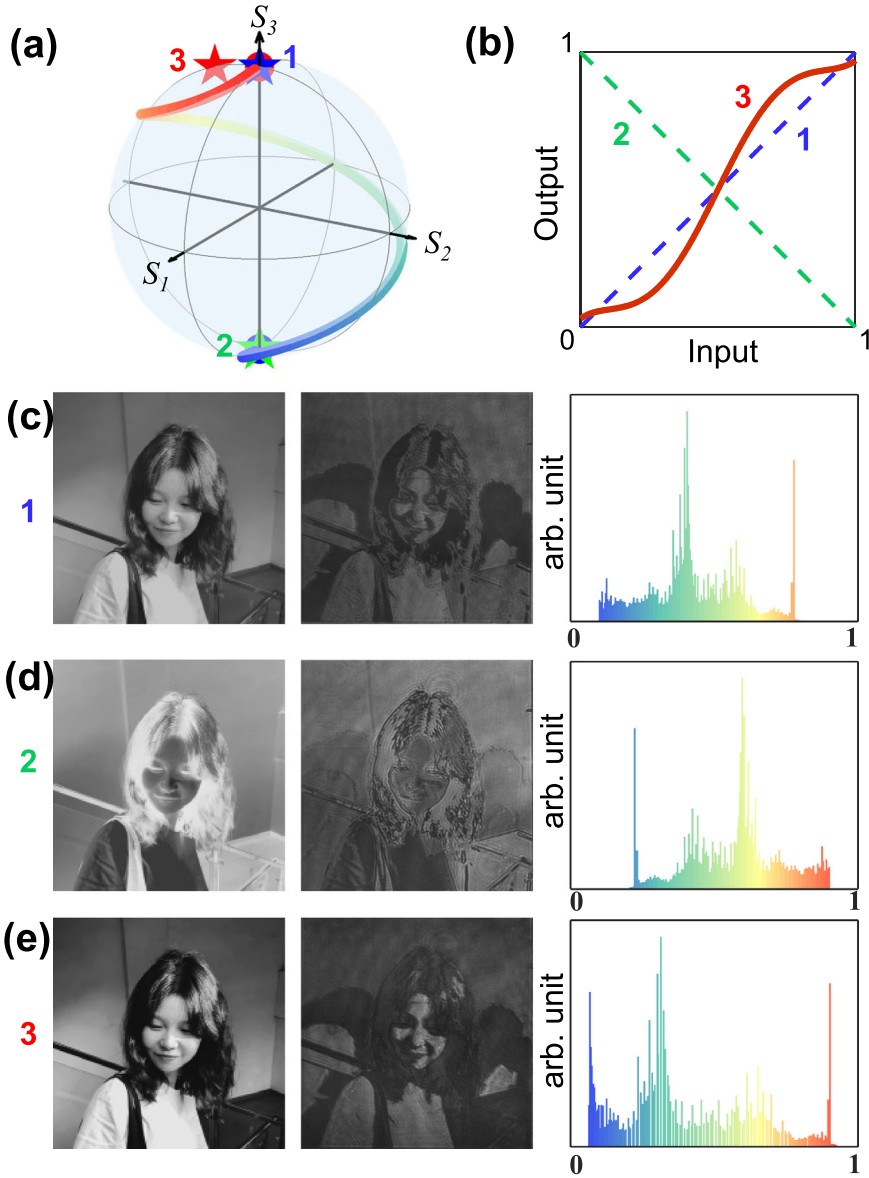

**Fig. 2 | Realized PS trajectory encoding metasurface for histogram stretching capability. a** The designed spiral trajectory governed by analytical function $\psi_l = 2\cos^2(\chi_l - \pi/4)$ under the circular basis on the PS; (**b**) Histogram transform curves at analyzer states 1: north pole ($\psi_A = 0^o$, $\chi_A = 45^o$), 2: south pole ($\psi_A = 0^o$, $\chi_A = -45^o$), and 3: ($\psi_A = 135^o$, $\chi_A = 35^o$); (**c**–**e**) Decoded images and their transformed histograms at the analyzer states 1, 2, 3 respectively.

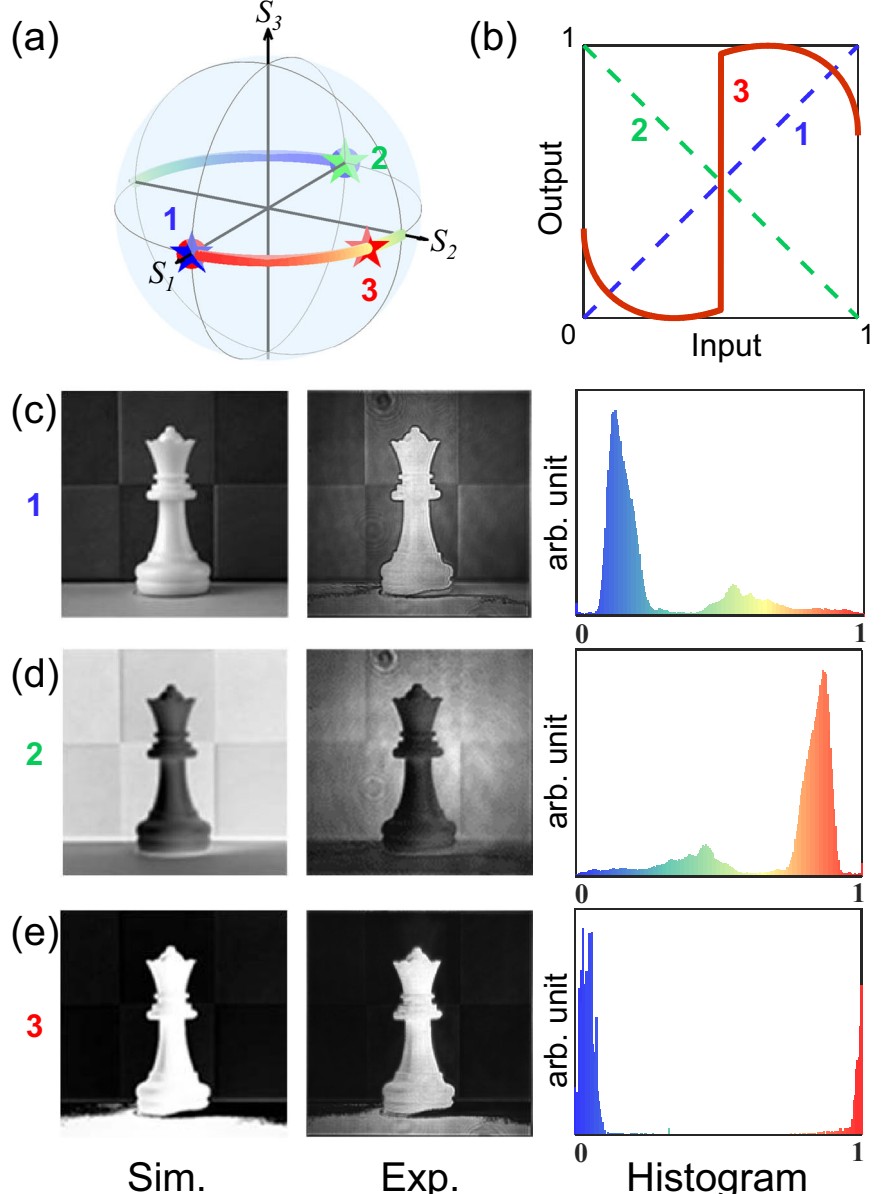

**Fig. 3 | Realized PS trajectory encoding metasurface for thresholding functionality. a** The designed pricewise trajectory $\psi_l = \{0^o\,(0 \leq \chi_l \leq 45^o)$; $90^o\,(-45^o \leq \chi_l \leq 0)\}$ under the linear basis of the PS. **b** Histogram transform curves at analyzer states 1: local north pole ($\psi_A = 0°$, $\chi_A = 0°$), 2: local south pole ($\psi_A = 90°$, $\chi_A = 0°$), and 3: ($\psi_A = 35°$, $\chi_A = 0°$); (**c**–**e**) Decoded images and their transformed histograms at the analyzer states 1, 2, 3 respectively.

As a final demonstration, we showcase how the higher-dimensional nature of the GML enables dual information channel encoding for arbitrary non-orthogonal polarizations, as shown in Fig. 4 and Supplementary Fig. 17. Here, instead of encoding the PS trajectory using an analytical function, we specify it using a grid constructed by a pair of arbitrarily aligned local PS. As shown in Fig. 4a, the grayscale of image A can be mapped onto the lines of latitude $\chi_l$ defined in one local PS system (PS I). Meanwhile, another local PS system (PS II) with a second set of lines of latitude $\chi'_l$ can be defined to independently encode a second image B (see Supplementary Note 2, Figs. 11 and 12 for the detailed encoding-decoding procedure). In the GML-based decoding process, the images A and B are decoded by analyzers placed on their individual local north poles, while at their orthogonal state (local south poles), the grayscale-inversed A and B images are analyzed. At other analyzer states beyond those four poles, mixed images of A and B are decoded. Remarkably, in contrast to conventional polarization multiplexing, here the dual information channels

do not need to be necessarily orthogonally polarized. When one information channel (local north pole on the PS I) is set, the polarization basis of the local PS II can vary on the equator of the PS I for perpendicular latitude grids (Fig. 4b), and even arbitrarily vary across the entire PS without the necessity of perpendicular grids (Supplementary Fig. 16 and Video 4). In this way, not only can we encrypt the information content, but also encrypt the location of information channels, providing enhanced security for advanced encryption applications. To experimentally demonstrate this prediction, we first realized a metasurface to encode the 'knight' image under right-handed circular polarization (RCP), and recoding the 'bishop' image in a linear polarization base with azimuthal angle $\psi'_{0l} = 22.5°$ (Fig. 4c). When the decoding analyzer is placed on the RCP state, the 'knight' image is efficiently decoded, and when the analyzer is placed in the linear polarization state with orientation angle 22.5°, the 'bishop' image is obtained. In order to then demonstrate the opportunity to choose the information channel, we designed another metasurface

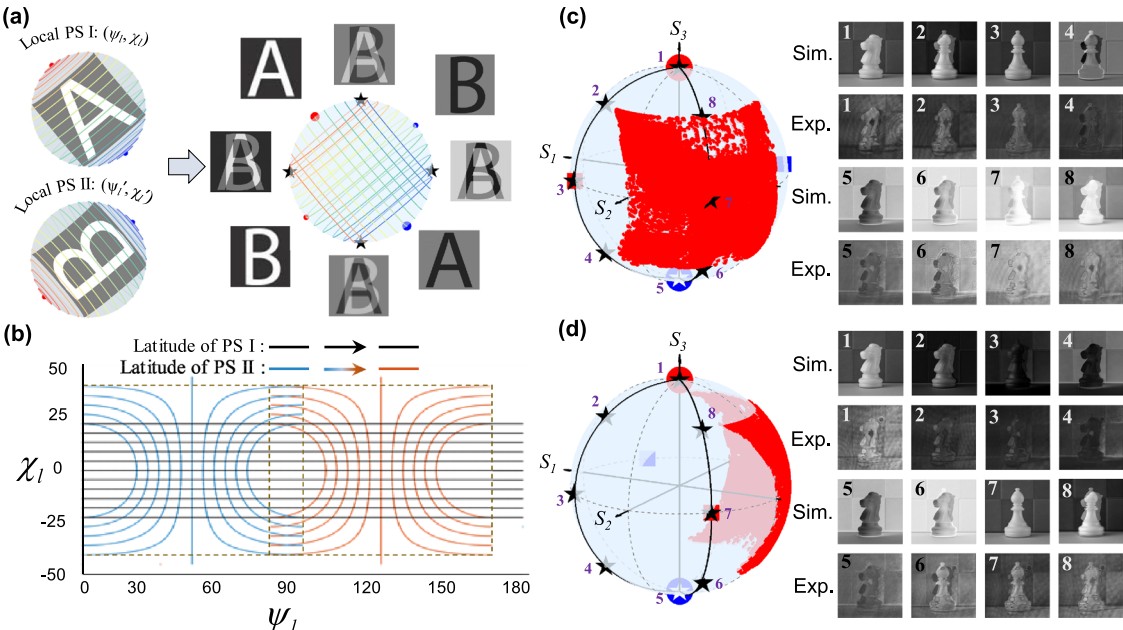

**Fig. 4 | PS trajectory encoding with arbitrarily aligned modulation grid of dual information channels. a** Left panel: the general design principle of the dual encoding images 'A' and 'B' at two local PS systems with orthogonal local latitude lines. Right panel: The decoded original 'A' and 'B' images are located at two local north poles (red dots), which are not orthogonal states. While at the two local south poles (blue dots), the grayscale-reversed individual images are analyzed. The mixed images of 'A' and 'B' or the mixing of their grayscale-reversed versions are decoded at other analyzing places (denoted as blue stars). **b** When local I is set, the PS II has infinite numbers of choices by moving its local north/south poles (together with the local coordinates $\psi_l$, $\chi_l$) around the equator of the PS I, suggesting another encryption knob for information multiplexing. **c** Dual information channel

encoding by setting PS I on the circular basis (circular dots) and PS II on the linear basis (square dots) with orientation angle $\psi_{0l}' = 22.5°$. The PS trajectory is represented by scattered red points on the square modulation area of the PS (left panel). The decoding analyzer sweeps two longitudes (black curves) containing 8 representative decoding positions (denoted as stars), the corresponding decoded images in simulation and experiment are shown in the right panel. **d** Keeping encoding polarization PS I the same as that of (**c**), but choose another PS II by shifting the north pole along the equator of the PS I to $\psi_{0l}'=67.5°$. By decoding the images with the same decoding path with positions 1-8, except for analyzing states 1 and 5 (where the original and grayscale-inversed image encoded at PSI appears), distinct images are decoded compared with the encoding process in (**c**).

recording the same 'knight' image under RCP polarization, but recording the second image in another linear polarization base with azimuthal angle $\psi_{0l}' = 67.5°$ (Fig. 4d). For the two different encoding bases, the decoded images at fixed analyzing paths (black curves) on the PS show different profile evolutions, which can build a library for the encryption signature associated with PS trajectory encoding. To further demonstrate arbitrary scenarios, we designed and realized another metasurface that encodes the dual information channels in two elliptical polarization bases without orthogonality (Supplementary Fig. 17). The overall captured images at the corresponding analyzer states indeed show the correct images, mixture of images and grayscale reversed images as expected from our theory.

## Discussion

We have demonstrated PS trajectory encoding of polarization information based on GML metasurfaces. The modulated PS trajectory between a pair of polarization basis poles can be set as an arbitrary analytical function or latitude grids constructed by two arbitrarily aligned local PS systems. By tailoring the modulation trajectory, polarization basis poles and analyzer polarizations, we realized a variety of analog image transformation operations based on metasurfaces, such as histogram stretching, thresholding and non-orthogonal channel encryption. Although in our current design the output field only provides the intensity variation information after decoding, the phase and amplitude degrees of freedom may be both engaged with polarization to encode additional non-orthogonal image channels in both near and far-field. Even though the polarization information encoded in the metasurface is fixed once it is fabricated, the GML metasurface can be combined with advanced computational imaging algorithms[33] to build reusable encryption and

anticounterfeiting schemes with enhanced information and security level for practical applications. The proposed technique of PS trajectory encoding based on GML greatly expands the applicability of polarization-encoding approaches already deployed in various technological platforms, and may find powerful applications in polarization optics, such as advanced LC displays, optical analog computation, information encryption and quantum polarization tomography.

## Methods

### Simulation of the PS trajectory encoding metasurfaces

In simulation, we calculate the Jones matrix of different structure parameters using a rigorous coupled-wave analysis (RCWA) solver[37–41] to set up the parameter lookup table. Based on the previous derivation, the target Jones matrix of the metasurfaces is obtained by using the coded polarization state. Finally, the parameters including the length, width, rotation angle, period and the height of meta-atom structure can be found by mapping the target Jones matrix with the lookup table.

### Fabrication of samples

The metasurfaces are designed by α-Si columns based on a SiO$_2$ substrate. Firstly, amorphous silicon (α-Si) thin films with a thickness of 658 nm were deposited on the fused quartz SiO$_2$ substrate by physical vapor deposition (PVD). After deposition, PMMA film was spin-coated and covered by PEDOT as a conductive layer. After the exposure process, the conducting layer was washed away when development. Dip the sample in hot acetone and clean it by ultrasonic to get the lift-off process done. Finally, the desired structure was transferred from Cr to silicon and the residual Cr was removed by cerium (IV) ammonium nitrate. The planar metasurfaces are composed of 1024 × 1024 periods (512 μm × 512 μm) with different coded polarizations and trajectories.

## Characterization of samples

A supercontinuum laser (Fianium-WL-SC480) is used as the near-infrared light source at 940 nm to display the encrypted hidden image information on the metasurfaces. In the experimental measurement, the circularly polarized incident light is generated by cascading a polarizer and a broadband quarter waveplate after the supercontinuous laser. Then, the circularly polarized laser beam is focused on the metasurfaces by an optical 4f system consisting of a 200 nm focal length and a 60-nm focal length lens. The resulting near-field image is amplified by a combination of an objective lens (NA = 0.15) and a tube lens. After going through another quarter-wave plate and an analyzer, the image was finally captured by the CCD camera. The PS location of the analyzer can be adjusted by rotating the second quarter-wave plate and the polarizer, yielding different decoded images on the CCD camera.

## Data availability

The data that supports the findings of this study are available from the corresponding authors upon request.

## Code availability

The code that supports the findings of this study are available from the corresponding authors upon request.

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

## Acknowledgements

This work was supported by the National Key Research and Development Program of China (2022YFB3607300 (Z.-L.D.), 2021YFB2802003 (X.L.)), National Natural Science Foundation of China (NSFC) (62075084 (Z.-L.D.) and 62325503 (X.L.)), Guangdong Basic and Applied Basic Research Foundation (2022B1515020004 (Z.-L.D), Guangzhou Science and Technology Program (2024A03J0465

(Z.-L.D.)), and the Air Force Office of Scientific Research (A.A.) and the Simons Foundation (A.A.).

## Author contributions

Z.-L.D. developed the concept. Z.-L.D. and M.-X.H. carried out the theoretical design and simulation of the PS trajectory encoding metasurfaces. M.-X.H, S.Q., and X.W. performed the experimental characterization and fabrication of the metasurface samples. Z.-L.D. drafted the manuscript with inputs from M.-X.H., A.O., X.L., and A.A. All authors contributed to discussions about the results in the manuscript.

## Competing interests

The authors declare no competing interests.
