## [Peer Review File · Nature Communications]

Poincaré Sphere Trajectory Encoding Metasurfaces Based on Generalized Malus' LawReviewer #1 (Remarks to the Author):

In "Generalized Malus' Law Metasurfaces for Poincaré Sphere Trajectory Encoding", Z-L Deng et al. report two developments (both of which are contained in the title):

- 1) A "Generalized Malus' Law" quantifying the intensity response of an arbitrary polarization state passing through an arbitrary polarization analyzer.
- 2) The use of this Generalized Malus' Law to produce a particular image in polarization to be seen when viewed through a particular polarization analyzer on the Poincaré Sphere ("encoding" in the authors' terminology). When the same polarization distribution is viewed through a different polarization analyzer (the "decoding" in the authors' terminology) a different image is seen, one that is potentially a highly nonlinear transformation of the original image.

As for (1), throughout the paper the authors imply that Malus' Law applies strictly to the \cos^2 intensity transfer characteristic governing linearly polarized light passing through a traditional linear polarizer. While Malus' Law may be presented this way in very introductory optics textbooks, light of any polarization passing through a polarization analyzer having any characteristic polarization state (lying anywhere on the Poincaré sphere) in fact behaves in accordance with a \cos^2 intensity-transfer relationship as the polarization state varies along a great circle path on the Poincaré sphere connecting the characteristic state of the polarizer with its orthogonal counterpart. This too is called Malus' Law - Malus' Law, both the traditional case with linearly polarized light and a linear polarizer and any polarization state/analyzer pair in general, is merely a consequence of the linear algebra that governs polarized light.

The authors claim that they have generalized Malus' Law, stating that their "introduced GML describes a universal rule for projection from one arbitrary elliptical polarization to another". What the authors are describing is already well-known, that being an inner product between two Jones vectors. It is not an intellectual contribution of this work. (I should also add that the authors' "Generalized Malus' Law" is not fully generalized, since the polarization state and the analyzer could both exhibit partial polarization; Malus' Law and the well-known linear algebra-based treatment of polarized light in terms of Stokes vectors can handle this case as well).

As for (2), the authors explore the freedom to analyze a particular spatial polarization distribution through different polarization analyzers. To describe this briefly, for a given input image, grayscale intensity is mapped along a given (possibly complex) trajectory along the Poincaré Sphere. A version of the image with this "encoded" polarization distribution is produced using a metasurface, but as the authors note this could also be done with other optical technologies such as liquid crystal displays. The encoding polarization distribution is designed such that when viewed through a particular polarization analyzer, the original image is displayed. However, viewed through different polarization analyzers, a different image is displayed, one that is possibly a highly nonlinear transformation of the original image which, as the authors point out, is performed in the analog domain by light. Fig. 3, in which an encoding path and analyzer can be chosen to essentially "threshold" an image, is a good example of this.

What the authors seem to imply is that their scheme is a way to use polarized light to perform analog computations, even those that are highly nonlinear (such as thresholding), with a passive optical system. However, this type of system is not generally useful for this purpose. Ideally, we would like a system that when presented with different images can perform the same transformation over and over again - this would be a fully analog device. In the authors' case, however, each new image requires a re-encoding in terms of a polarization distribution, which then requires the design and fabrication of a metasurface. Only when a polarization analyzer acts upon this new spatial polarization distribution will the transformed image be obtained. In other words, the encoding process is a part of the information processing system proposed by the authors. The encoding is done digitally in order to compute the

correct polarization distribution. In other words, this approach doesn't live up to the vision laid out in the paper, in that it is not an all-analog image processing system. In order to obtain highly non-linear transformations in the light domain, the encoding itself must be a highly nonlinear computation first performed in the digital domain. For instance, if we use thresholding as in Fig. 3 as an example, the encoding procedure must already apply a thresholding procedure on the input image intensity digitally during the transformation to a Poincaré path. The "hard" part of the operation is done on an electronic digital computer; what we see in the optics domain is just a view of what has already been computed using more traditional means. I thus do not see an advantage to the authors' approach.

For these reasons, I cannot recommend this work for publication in Nature Communications.

Reviewer #2 (Remarks to the Author):

This paper introduces a generalized version of Malus' law, which traditionally governs the behavior of plane-polarized light passing through a linear polarizer. This law is crucial for polarization optics used in various technologies like liquid crystal displays and optical encryption. However, it's limited to linear polarization on the Poincaré sphere. The study showcases how metasurfaces, a new technology, can extend the application of Malus' law to arbitrary polarization pairs across the entire Poincaré sphere. By controlling polarization trajectories through mathematical functions or alignment with specific grids, this approach allows for custom polarization mappings. Consequently, it enables novel all-optical image processing functions like histogram stretching, thresholding, and image encryption within non-orthogonal regions on the Poincaré sphere. This innovation merges polarization optics with analog optical processing, offering exciting possibilities for imaging and optical applications in both quantum and classical domains. In summary, the paper significantly expands polarization optics' capabilities and its applications in optical image manipulation and processing.

Here are few minor points that the authors should clarify in the manuscript:

1. The authors emphasize that while Malus' law serves as a mathematical basis for explaining intensity changes when two linear polarization filters are combined, it doesn't account for elliptical or circular polarization characteristics. In contrast, there are alternative formulations like the Stokes general equation that accurately describe intensity changes when light passes through both a linear polarization filter and a retarder. This equation would be similar to the general Malus law shown here. The approach proposed in this paper shares similarities with the latter method, and it would be valuable to compare it to such methods since they offer a more comprehensive framework for modeling polarization states on the Poincaré sphere. See for example the discussion in chapter 6 Denis Goldstein book on Polarization Optics. The main concern is that the term "general Malus law" is the same (or similar) to the Stokes equation that is already used in the polarization community.
2. Previous studies have explored the capture of the full Stokes vector using optimally designed retarders, as evidenced by references such as 10.1088/2040-8978/18/5/055702 and 10.1364/OL.44.002927. These works share a similar approach of encoding incoming polarization states with a set of retarders and linear polarization to achieve optimal reconstruction (decoding) in terms of signal-to-noise ratio (SNR). In the presented work, this concept is extended to metasurfaces, which represents a highly innovative approach. However, it is essential for the paper to provide a comparison with these conventional methods, outlining their respective advantages and disadvantages.
3. The paper presents a compelling array of experimental data, offering numerous test images that effectively highlight the capabilities of their approach. Nonetheless, the paper falls short in providing concrete, quantifiable performance metrics. It is crucial that the paper incorporates measurements capable of quantifying the accuracy or error in their reconstruction (encoding-decoding) process for a range of input states.

Reviewer #3 (Remarks to the Author):

This is nice fundamental work that generalizes Malus' law to project from one arbitrary elliptical polarization to another. An elegant formulation has been derived to mathematically express this generalized Malus' law, as shown in equations (1) and (2) in the main text. In this formulation, it was revealed that the local azimuthal parameter ψ_l can be used as a degree of freedom for polarization encoding following an arbitrary trajectory on the Poincare sphere. This theory has then been experimentally verified using a dedicated metasurface to showcase several analogue image processing experiments at lightwave frequencies.

In my opinion this work is excellent and of fundamental nature.

Moreover, the supporting experimental results are convincing. If I have some suggestion is to try and explain better the experimental process of encoding/decoding. I had hard time trying to understand this, despite the details provided in the Supplementary material (e.g. Figures S9,S10). How is the pixel-by-pixel encoding process implemented in practice? I suggest summarizing what is shown in S9,S10 with a short but descriptive text in the main body of the paper where the reference is made to Figures S9, S10. Perhaps also a figure in the Supplementary, like S11, which actually explains how the object is involved would be useful.

Re. NCOMMS-23-47496

A detailed point-by-point response to all the reviewers' questions is provided as follows. The original remarks of the reviewers are reproduced in **black**; our responses are provided in **blue**.

Response to Reviewer #1

GENERAL COMMENTS

In "Generalized Malus' Law Metasurfaces for Poincaré Sphere Trajectory Encoding", Z-L Deng et al. report two developments (both of which are contained in the title):

- 1) A "Generalized Malus' Law" quantifying the intensity response of an arbitrary polarization state passing through an arbitrary polarization analyzer.
- 2) The use of this Generalized Malus' Law to produce a particular image in polarization to be seen when viewed through a particular polarization analyzer on the Poincaré Sphere ("encoding" in the authors' terminology). When the same polarization distribution is viewed through a different polarization analyzer (the "decoding" in the authors' terminology) a different image is seen, one that is potentially a highly nonlinear transformation of the original image.

As for (1), throughout the paper the authors imply that Malus' Law applies strictly to the \cos^2 intensity transfer characteristic governing linearly polarized light passing through a traditional linear polarizer. While Malus' Law may be presented this way in very introductory optics textbooks, light of any polarization passing through a polarization analyzer having any characteristic polarization state (lying anywhere on the Poincaré sphere) in fact behaves in accordance with a \cos^2 intensity-transfer relationship as the polarization state varies along a great circle path on the Poincaré sphere connecting the characteristic state of the polarizer with its orthogonal counterpart. This too is called Malus' Law - Malus' Law, both the traditional case with linearly polarized light and a linear polarizer and any polarization state/analyzer pair in

general, is merely a consequence of the linear algebra that governs polarized light.

The authors claim that they have generalized Malus' Law, stating that their "introduced GML describes a universal rule for projection from one arbitrary elliptical polarization to another". What the authors are describing is already well-known, that being an inner product between two Jones vectors. It is not an intellectual contribution of this work. (I should also add that the authors' "Generalized Malus' Law" is not fully generalized, since the polarization state and the analyzer could both exhibit partial polarization; Malus' Law and the well-known linear algebra-based treatment of polarized light in terms of Stokes vectors can handle this case as well).

Response: We thank the reviewer very much for the thorough and careful reading of our manuscript and the pertinent summary of the development of our work.

Indeed, in many introductory optics textbooks, Malus' Law is usually presented as the intensity-transfer relationship between input linear polarizations and linear polarizer, and the relationship between the generalized arbitrary input polarization and analyzer could be deduced with some linear algebra that governs polarized light. We emphasize that, the contribution of our work is mainly the usage of analytical forms of GML for the proposed Poincaré sphere trajectory encoding by exploiting the hidden dimension in the polarization mapping relation. This approach fully releases the additional dimension for brand new information encoding that is not feasible based on previous approaches. To remove the emphasis on the introduction or proposal of the GML concept, we have revised the title as “Poincaré Sphere Trajectory Encoding Metasurfaces with Generalized Malus’ Law”. In addition, we have rephrased the claim of “our introduced GML” in the abstract and introduction part more precisely as, “introduce a Poincaré sphere (PS) trajectory encoding scheme for versatile polarization information encoding by employing a GML”, which truly reflects the innovation of this work on the usage of GML to the polarization encoding methodology with revealed additional freedom. We hope that, in this way, it clearly states the contribution of our work.

It should be pointed out that although the Malus law together with some linear algebra-based treatment of light polarization could handle this case in principle, our derived analytical expression of GML can provide further comprehensiveness and advances in the following two factors. First, our approach is more straightforward than the pure mathematic procedure of the Stocks vector treatment to gain the physical insights into the universal polarization projection phenomenon. Even the great circle picture manifests a graphical projection relationship, however, it does not clearly indicate quantitative relations in terms of full Poincare sphere parameters, which are clearly represented by our constructed transformation between local and global Poincare sphere framework. Second, although the standard mathematic procedure can calculate the output Stocks vector by a given Mueller matrix and input Stocks vector, our approach demonstrates superiority in providing analytical formulas of the polarization projected intensity in any arbitrary cases. One should calculate the projected results one by one for given input polarizations, therefore the overall physical picture is not clear. Instead, our derivation of the GML expression based on the affine transformation between the local and the global PS system gives straightforward and clear physical picture of the projection relation, and readily provides the analytic expression of the relation in both local and global forms. In particular, the elegant local form reveals the additional dimension in the mapping relation that is a totally uncharted degree of freedom in previous polarization encoding approaches. It enables us to freely encode polarization trajectory and imperceptible information in terms of both information content and location as our work demonstrated. It should be mentioned that this insight could hardly be observed by the routine mathematic procedure with Mueller matrix and Stocks vector treatment.

To address the reviewer's concern about the lack of partial polarization consideration in our GML, we have added significant further work to extend the polarization projection relation in the most general case, in which both the incident light and the analyzer could be partially polarized or unpolarized. In the Poincare sphere representation, it means that, both the input polarization and the analyzer polarization could be located on the *solid* PS including not just its surface but also its interior. To

derive the GML including partial polarized and unpolarized light, we set the polarization parameters (azimuth, ellipticity, DoP) of the incident light and the analyzer as (ψ, χ, p) and (ψ_0, χ_0, p_0) , respectively. Where the addition parameter DoP is the degree of polarization that is represented by the radius coordinate in the solid Poincare sphere. To simplify the derivation procedure, we first deal with the projection relation in the local Poincare sphere system based on the polarization basis of analyzer. The local polarization parameters of the input light and the analyzer are denoted as (ψ_l, χ_l, p) and $(0^\circ, 45^\circ, p_0)$, respectively. Note that, the DoP parameters remain the same in both local and global Poincare sphere system, because only rotation transformation is performed between them; the local polarization coordinate of the analyzer located on the RCP pole as it is in its own PS system.

The partial polarizer could be characterized by the Jones matrix in its local basis as,

$$J_p = \begin{pmatrix} 1 & 0 \\ 0 & c_0 \end{pmatrix}, \quad (1)$$

which represent that, its allowed state $|\alpha_0(\psi_0, \chi_0)\rangle$ has a transmission coefficient of unity, while the orthogonal state $|\alpha^\perp_0\rangle$ also have a finite transmission coefficient c_0 , which connects the DoP of the polarizer by [Ref. to Phys. Rev. Lett. 130(12), 123801 (2023), (Ref. 34 in the manuscript)],

$$p_0 = \frac{1-|c_0|^2}{1+|c_0|^2}, \text{ or } |c_0|^2 = \frac{1-p_0}{1+p_0}, \quad (2)$$

Under the local system, we first divide the input partial polarized light into two portions: 1. A portion of unpolarized light power weighted by $(1-p)$; 2. A portion of polarized light with polarization parameters (ψ_l, χ_l) weighted by p .

1. When a beam of unpolarized light with unitary power passes through the partial polarizer, the output light become a partial polarized light composed of a portion of unpolarized light with power,

$$P_{1u} = |c_0|^2, \quad (3)$$

and a portion of fully polarized light with power,

$$P_{1p} = (1 - |c_0|^2)/2, \quad (4)$$

Therefore, the total output power of the projected partial polarized light is,

$$P_{1total} = P_{1u} + P_{1p} = (1 + |c_0|^2)/2, \quad (5)$$

Substituting Eq. (2) in to Eq. (5), we obtain,

$$P_{1total} = 1/(1 + p_0), \quad (6)$$

Therefore, $(1-p)$ portion of unpolarized light in the incident light will produce output

intensity of,

$$P_1 = (1 - p)/(1 + p_0). \quad (7)$$

2. When the full polarized light $|\alpha(\psi_l, \chi_l)\rangle$ with unity power, is incident upon the same partially polarizing element, the resulting light intensity, denoted as I_p , can be obtained by the Jones matrix calculation in the local circular polarization basis.

$$\alpha_p = J_p |\alpha_l(\psi_l, \chi_l)\rangle = \begin{pmatrix} 1 & 0 \\ 0 & c_0 \end{pmatrix} \exp(i\Phi) \begin{bmatrix} \cos\left(\frac{\pi}{4} - \chi_l\right) e^{i\psi_l} \\ \sin\left(\frac{\pi}{4} - \chi_l\right) e^{-i\psi_l} \end{bmatrix} = \exp(i\Phi) \begin{pmatrix} \cos(\pi/4 - \chi_l) e^{i\psi_l} \\ c_0 \sin(\pi/4 - \chi_l) e^{-i\psi_l} \end{pmatrix}, \quad (8)$$

Therefore, the transmitted intensity can be written as

$$p_{2total} = \cos^2(\pi/4 - \chi_l) + |c_0|^2 \sin^2(\pi/4 - \chi_l), \quad (9)$$

Note that, Eq. (S35) represents the projection from a polarized light to a partial polarizer, which was consistent with some previous work on the polarization projection involving imperfect polarizers [Ref. to, for example, J. Opt. 21(10), 105001 (2019) (Ref. 15 in the revised supplementary material)], which is only a *small subset* of our present GML at the most general case.

Therefore, p portion of fully polarized light will produce output light with power,

$$p_2 = p * p_{2total} = p \left[\cos^2\left(\frac{\pi}{4} - \chi_l\right) + \frac{1-p_0}{1+p_0} \sin^2\left(\frac{\pi}{4} - \chi_l\right) \right], \quad (10)$$

The total output power from the partial polarized light to the partial polarizer in the local Poincare sphere system yields,

$$I_{out} = A_0^2(p_1 + p_2) = \frac{1-p}{1+p_0} + p \left[\cos^2\left(\frac{\pi}{4} - \chi_l\right) + \frac{1-p_0}{1+p_0} \sin^2\left(\frac{\pi}{4} - \chi_l\right) \right] = A_0^2 \frac{1-pp_0}{1+p_0} + A_0^2 \frac{2pp_0}{1+p_0} \cos^2\left(\frac{\pi}{4} - \chi_l\right), \quad (11)$$

where, A_0 is the amplitude of the incident light. This is the local form of the *solid* PS GML. To transfer the local form to the global form, affine transformation between the local and global coordinates, we finally obtain the global form of the *solid* PS GML as,

$$I_{out} = A_0^2 \frac{1-pp_0}{1+p_0} + A_0^2 \frac{pp_0}{1+p_0} [\cos 2\chi_0 \cos 2\chi \cos(2\psi - 2\psi_0) + \sin 2\chi_0 \sin 2\chi + 1], \quad (12)$$

When both p and p_0 equal 1, it collapses to the scenario where fully polarized light is incident upon a perfectly polarized element,

$$I_{out} = \frac{1}{2} A_0^2 [\cos 2\chi_0 \cos 2\chi \cos(2\psi - 2\psi_0) + \sin 2\chi_0 \sin 2\chi + 1], \quad (13)$$

Which is the GML on the *surface* PS, agreeing with Eq. (2) in our maintext.

The above further work has been added in the revised manuscript, please refer to Paragraph 4, Page 3 of the maintext, and Notes 1.2 of the supplementary material.

As for (2), the authors explore the freedom to analyze a particular spatial polarization distribution through different polarization analyzers. To describe this briefly, for a given input image, grayscale intensity is mapped along a given (possibly complex) trajectory along the Poincaré Sphere. A version of the image with this "encoded" polarization distribution is produced using a metasurface, but as the authors note this could also be done with other optical technologies such as liquid crystal displays. The encoding polarization distribution is designed such that when viewed through a particular polarization analyzer, the original image is displayed. However, viewed through different polarization analyzers, a different image is displayed, one that is possibly a highly nonlinear transformation of the original image which, as the authors point out, is performed in the analog domain by light. Fig. 3, in which an encoding path and analyzer can be chosen to essentially "threshold" an image, is a good example of this.

What the authors seem to imply is that their scheme is a way to use polarized light to perform analog computations, even those that are highly nonlinear (such as thresholding), with a passive optical system. However, this type of system is not generally useful for this purpose. Ideally, we would like a system that when presented with different images can perform the same transformation over and over again - this would be a fully analog device. In the authors' case, however, each new image requires a re-encoding in terms of a polarization distribution, which then requires the design and fabrication of a metasurface. Only when a polarization analyzer acts upon this new spatial polarization distribution will the transformed image be obtained. In other words, the encoding process is a part of the information processing system proposed by the authors. The encoding is done digitally in order to compute the correct polarization distribution. In other words, this approach doesn't live up to the vision laid out in the paper, in that it is not an all-analog image processing system. In order to obtain highly non-linear transformations in the light domain, the encoding itself must be a highly nonlinear computation first performed in the digital domain. For instance, if we use thresholding as in Fig. 3 as an example, the encoding procedure must already apply a thresholding procedure on the input image intensity digitally during the transformation

to a Poincaré path. The "hard" part of the operation is done on an electronic digital computer; what we see in the optics domain is just a view of what has already been computed using more traditional means. I thus do not see an advantage to the authors' approach.

Response: We thank the reviewer for the in-depth understanding of our work. Indeed, the most general case for the optical analog computation should perform the same computation operation for multiple input images over and over again to be a fully analog device. For our case, both the input image and the computation operation are encoded in the metasurface. New image requires a re-encoding in terms of a polarization distribution, which then requires the design and fabrication of a metasurface. Because, typically reconfigurable metasurface is difficult to be realized due to the dynamic limitation of the constituent materials at the pixelated level, which is the biggest challenge in the metasurface field. Our proposed Poincare sphere trajectory encoding approach is a general polarization information encoding methodology that could perform arbitrary nonlinear transformations of encoded polarization image in the metasurface. The general encoding methodology is not restricted to some special image processing operations such as the thresholding and the histogram stretching. To remove the reviewer's concern that our approach is not general useful for the all-analog image processing purpose, we do not claim the direct connection of our work to the image processing system, but just emphasize on the concept of polarization information encoding, which correctly describes the function of our approach. In the revised manuscript, we have rephrased our claim to not directly connect our proposed Poincare sphere trajectory encoding approach to the image processing system, but discuss it in scope of polarization information encoding, please refer to the first paragraph in page 1 and 2 in the revised manuscript.

Nevertheless, we should point out that, even though our proposed approach is not directly related to the all-analog image processing system that requires the capability to process various input images over and over again in the same way without change the structure of the metasurface, the Poincare sphere trajectory encoding technique as a

general polarization encoding approach still manifests significant advantages compared with conventional polarization encoding techniques. As conventional bulky polarization optical elements are all based on linear birefringence and diattenuation effects, the polarizers and waveplates are built on the linear polarization basis, which could only filter linear polarized light and produce retardance between orthogonal linear polarization pairs, respectively. Because of this, previous polarization encoding in both conventional and metasurface way always relies on the polarization projection from one arbitrary polarization onto a linear polarization state. Many polarization optics related applications, such as liquid crystal displaying, 3D glasses, quantum polarization entanglement construction are all based on the linear polarization – intensity mapping. Even dealing with complex polarization beyond linear polarization, the intensity mappings are always transformed to the linear polarization basis, which is still based on the framework of conventional Malus' law describing a 1D (a single parameter with respect to the polarization angle) projection relation. For example, recently reported Malus metasurfaces have the capability of polarization image encoding in a spatial-varying way with ultra-high resolutions [Light Sci. & Appl. 7, 17129 (2018), Light Sci. & Appl. 9(1), 101 (2020), Sci. Adv. 7(21), eabg0363 (2021), et. al. (Ref. 28-33 in the maintext)], facilitating advanced encryption and multiplexing applications. However, all of them are still based on the original version of Malus law that project one linear polarization to another linear polarization. As the original Malus law only describe the polarization projection relation in a 1D space, the encoding diversity and flexibility is largely limited.

The PS trajectory encoding based on the generalized Malus' law, actually **promotes the projection dimension from one to two**. The dimension promotion significantly opens new windows for more possibilities. The additional dimension in the mapping relation is a totally uncharted degree of freedom and can be used to freely encode polarization trajectory and imperceptible information in terms of both information content and location.

To clearly illustrate the advantage of the present PS trajectory encoding with the GML framework, we make a comparison between the conventional polarization

encoding with Malus' law and the PS trajectory encoding with the GML on the PS, as shown in Fig. R1. The polarization base poles (red and blue spheres) which represent the polarization bases for the pixel-by-pixel polarization encoding of the incident light beam, the polarization encoding trajectory (color curve) and the analyzer polarization state (denoted by stars) are all restricted on the equator of the PS [Fig. R1 (a)], the encoding flexibility is largely restricted. That is, as the polarization projection relation in Malus' law is one-dimensional, the polarization-intensity mapping is a one-to-one mapping. When the polarization encoding of an image is completed, the decoded (analyzed) images at all possible PS positions are determinate, therefore, the encoding-decoding relationship is univocal, and many complex encryption or steganography process cannot be established.

On the other hand, for the PS trajectory encoding based on GML [Fig. R1 (b-d)], the modulation trajectory between the encoding base poles is no longer restricted to the great circle path connecting those two poles, but be set as an arbitrary function between the local ellipticity χ_l and local azimuthal ψ_l parameters: $\psi_l=f(\chi_l)$. In addition, the polarization encoding base poles are no longer restricted on the equator of the PS, but can span over all the PS [Fig. R1 (c)]. Note that the modulation trajectory function can be any analytical function, piecewise function, or even a discrete data list, as long as we can uniquely address ψ_l at a given χ_l . In this way, an additional degree of freedom beyond the information content itself, namely, the modulation trajectory and as a result the disclosure position of the encoded information could be unprecedentedly encrypted. This way largely increases the diversity of polarization encoding. Furthermore, the decoding analyzer (black star) can also be released from the equator of the PS to all possible locations on the PS surface, further increasing the capacity of the encoding-decoding procedure [Fig. R1 (d)].

In terms of the functionality, the conventional polarization encoding based on Malus law only has limited input-output transformation relations, as shown in Fig. R2 (a), it is not flexible enough to design some meaningful image transformation operations, as the simulation results in Fig. R2 (a) and the experimental results in Ref.

Fig. R1. Evolution of polarization encoding from 1D projection based on Malus' Law to Poincare sphere trajectory encoding based on Generalized Malus' law. The red/blue dots are the north/south poles of the local Poincare sphere system, representing the polarizer allowed and stopped states, respectively. The black stars represent the analyzer locations to decode the image. The colored curves represent the modulation trajectory, where the color denotes the modulated intensity on the corresponding location of the Poincare sphere. (a) For polarization encoding with Malus' law, both the encoding trajectory and decoding analyzer locations are restricted to the equator of the Poincare sphere. (b) For Poincare sphere trajectory encoding with generalized Malus' law, the modulation trajectory can be expanded from the equator of the Poincaré sphere to an arbitrary path connecting the polarization basis poles as denoted by the red and blue dots. (c) The polarization basis pole positions can also be expanded from the equator of the Poincaré sphere to any arbitrary positions on the Poincaré sphere. (d) The decoding analyzer positions could also be distributed to any loci of the Poincaré sphere rather than being restricted on the equator of the Poincaré sphere. (d) shows the most general case of the Poincaré sphere trajectory encoding and decoding process with extended possibility of modulation trajectory, encoding polarization bases and decoding analyzer positions.

Fig. R2. Comparison of the diversity of input-output relation and output images between (a) polarization encoding based on conventional Malus' law and (b-e) PS trajectory encoding based on generalized Malus' law.

In summary, our work discovers the hidden dimension in the polarization mapping relation by employing the generalized Malus law spanning over the entire PS, and fully exploits the additional dimension for a brand-new information encoding approach that is not feasible based on previous reports, which may bring new opportunities for LC applications, novel polarization optical element designs and even quantum polarization state entanglement.

Response to Reviewer #2

GENERAL COMMENTS

This paper introduces a generalized version of Malus' law, which traditionally governs the behavior of plane-polarized light passing through a linear polarizer. This law is

crucial for polarization optics used in various technologies like liquid crystal displays and optical encryption. However, it's limited to linear polarization on the Poincaré sphere. The study showcases how metasurfaces, a new technology, can extend the application of Malus' law to arbitrary polarization pairs across the entire Poincaré sphere. By controlling polarization trajectories through mathematical functions or alignment with specific grids, this approach allows for custom polarization mappings. Consequently, it enables novel all-optical image processing functions like histogram stretching, thresholding, and image encryption within non-orthogonal regions on the Poincaré sphere. This innovation merges polarization optics with analog optical processing, offering exciting possibilities for imaging and optical applications in both quantum and classical domains. In summary, the paper significantly expands polarization optics' capabilities and its applications in optical image manipulation and processing.

Here are few minor points that the authors should clarify in the manuscript.

Response: We thank the reviewer for endorsing that our work offers exciting possibilities for imaging and optical applications in both quantum and classical domains, and significantly expands polarization optics' capabilities and its applications in optical image manipulation and processing. In the following, we will reply to your comments point-by-point.

Comment 1) The authors emphasize that while Malus' law serves as a mathematical basis for explaining intensity changes when two linear polarization filters are combined, it doesn't account for elliptical or circular polarization characteristics. In contrast, there are alternative formulations like the Stokes general equation that accurately describe intensity changes when light passes through both a linear polarization filter and a retarder. This equation would be similar to the general Malus law shown here. The approach proposed in this paper shares similarities with the latter method, and it would be valuable to compare it to such methods since they offer a more comprehensive framework for modeling polarization states on the Poincaré sphere. See for example the

discussion in chapter 6 Denis Goldstein book on Polarization Optics. The main concern is that the term “general Malus law” is the same (or similar) to the Stokes equation that is already used in the polarization community.

Response: We thank the reviewer for this important comment. We definitely agree that, the general Mueller Matrix and the Stocks equation offer a more comprehensive framework for modeling polarization states on the Poincaré sphere. As Mueller Matrix describes the relation between Stock parameters of arbitrary input light and output light, as long as we build the corresponding Mueller Matrix of that can filter an arbitrary elliptical polarization state, and set the input Stock parameters as another arbitrary elliptical polarization state, the final output intensity could indeed be obtained by calculating those equations. Our derived GML expression can provide further comprehensiveness and advances in the following two factors.

1) Compared to the cumbersome overall mathematic procedure, our approach can provide the explicit polarization projection expression in terms of input azimuth, ellipticity and DoP parameters (ψ, χ, p) and the analyzer orientation and ellipticity parameters (ψ_0, χ_0, p_0) . Our mentioned generalized Malus law is derived by constructing the local PS in terms of the analyzer’s allowed polarization sate and its orthogonal state, and derives the polarization projection law at the local system with a simple form of,

$$I_{out} = A_0^2 \frac{1-pp_0}{1+p_0} + A_0^2 \frac{2pp_0}{1+p_0} \cos^2 \left(\frac{\pi}{4} - \chi_l \right), \quad (14)$$

where, χ_l is the local ellipticity parameter of the input polarization. With the help of affine transformation between the local Poincare sphere system and the global Poincare sphere system, explicit expression of the universal polarization projection expression in terms of input orientation and ellipticity parameters (ψ, χ, p) and the analyzer orientation and ellipticity parameters (ψ_0, χ_0, p_0) is readily arrived at as follows,

$$I_{out} = A_0^2 \frac{1-pp_0}{1+p_0} + A_0^2 \frac{pp_0}{1+p_0} [\cos 2\chi_0 \cos 2\chi \cos(2\psi - 2\psi_0) + \sin 2\chi_0 \sin 2\chi + 1]. \quad (15)$$

Here, the GML represent the most general case of polarization projection, where both the incident light and polarizer could be placed on arbitrary position of the *solid* PS.

2). The explicit projection relation by our approach deliveries a clear physical picture and reveals hidden degree of freedom for polarization information encoding. The local

form of GML implies that, there is an uncharted degree of freedom in the projection, which can be arbitrary assigned. Note that, this insight could not be observed by only considering the overall input and output relations governed by the Mueller matrix and the Stocks equation. With both the local and global form with explicit expressions of the Generalized Malus law can be directly used to encode and decode the polarization information, respectively, which provides deeper physical insights than that described in the textbook.

Following the reviewer's suggestion, we have also carefully read Benis Goldstein's book on Polarization Optics, especially, the chapter 6. It comprehensively discussed the Mueller Matrices for a variety of conventional polarization optical elements, such as linear polarizer, linear retarder, and the generation of elliptical polarized light by cascading a polarizer and a retarder. Although the Mueller Matrix describes the relation between Stock parameters of arbitrary input light and output light, the most general polarization projection relation is not explicitly derived yet. As conventionally, people usually deal with the Mueller matrices and Stocks equations of existing bulky polarization optical elements, which are far less flexible and diverse than the metasurface polarization elements. The most general case the book has discussed is the cascading system of a rotated polarizer and a fixed retarder as shown in Fig. R3. The forms of the polarizer and the retarder is not general, as the orientation of the retarder is set, and the purpose of the Mueller matrix is to generate an arbitrary elliptically polarization state, which fails to describe the projection relation between one arbitrary elliptical polarization and another arbitrary elliptical polarization.

The Mueller matrix of a rotated ideal linear polarizer is

$$\mathbf{M}_p(2\theta) = \frac{1}{2} \begin{pmatrix} 1 & \cos 2\theta & \sin 2\theta & 0 \\ \cos 2\theta & \cos^2 2\theta & \sin 2\theta \cos 2\theta & 0 \\ \sin 2\theta & \sin 2\theta \cos 2\theta & \sin^2 2\theta & 0 \\ 0 & 0 & 0 & 0 \end{pmatrix}. \quad (6.106)$$

FIGURE 6.7 The generation of elliptically polarized light.

Fig. R3. Mueller matrix representation of cascaded polarizer and retarder in the textbook.

Related classification and additional work have been added in our revised manuscript, please see lines 7-17 in the 2nd paragraph on page 3 in the revised manuscript, and Supplementary Notes 1.2 in the supplementary materials.

Comment 2) Previous studies have explored the capture of the full Stokes vector using optimally designed retarders, as evidenced by references such as 10.1088/2040-8978/18/5/055702 and 10.1364/OL.44.002927. These works share a similar approach of encoding incoming polarization states with a set of retarders and linear polarization to achieve optimal reconstruction (decoding) in terms of signal-to-noise ratio (SNR). In the presented work, this concept is extended to metasurfaces, which represents a highly innovative approach. However, it is essential for the paper to provide a comparison with these conventional methods, outlining their respective advantages and disadvantages.

Response: Thank the reviewer for providing this valuable suggestion. We have carefully read those papers, which provide approaches of encoding incoming polarization states with a set of retarders and linear polarizers to achieve optimal decoding, indeed providing a very excellent strategy. We can make the comparison of our proposed approach and those conventional methods in the following aspects:

1. Our approaches use single-layered metasurface that modulate the polarization

information in a pixel-by-pixel spatially-varying way; while the conventional methods cascade multiple bulky retarders and polarizers to modulate the polarization of the whole optical beam.

2. The application scenarios are different. Our encoding approaches mainly aim for the generation of some particular polarization distributions and a variety of image transformation relations; while the conventional encoding approaches pointed out by the two papers aim for the measurement of polarizations of the incoming beam. With this respect, the advantage of our approach is that, the polarization distribution and the image transformation relation after the polarization encoding could be much more diverse and flexible than conventional approaches. As for the disadvantage, our approach is not good at the precise measurement of the polarization state of an incoming light field with versatile polarization profiles, especially when many kinds of noises such as Poisson noise and Gaussian noise exist in the incoming light. Nevertheless, as the purpose of our approach is to manipulate rather than detect specialized polarization field, therefore the pursuit of diversity is preferred to the precision requirement.

3. Most importantly, our encoding approaches innovatively introduces the Poincare sphere trajectory engineering, which is totally out of scope of previous conventional approaches. In our approach, the polarization mapping could be engineered by an arbitrary trajectory lies on the Poincare sphere, providing much more possibilities.

To address this point, we have added the comparison of our encoding approaches and the conventional methods suggested in the reviewer mentioned papers in our revised manuscript. Please refer to lines 6-10, last paragraph, page 4, and added reference Ref. 35 and Ref. 36.

Comment 3) The paper presents a compelling array of experimental data, offering numerous test images that effectively highlight the capabilities of their approach. Nonetheless, the paper falls short in providing concrete, quantifiable performance metrics. It is crucial that the paper incorporates measurements capable of quantifying the accuracy or error in their reconstruction (encoding-decoding) process for a range of input states.

Response: Thank the reviewer very much for endorsing that our paper presents a compelling array of experimental data. To quantify the accuracy and error in our reconstruction process for a range of input states, we have added further experimental data analysis of the encoding-decoding procedure. As shown in Fig. R4 (Fig. S14 in the revised supplementary material), to evaluate the experimental performance of the modulated transform function, we redesigned two linearly gradient grayscale images modulated by the *S*-shape and thresholding curves, respectively. We sample points on the captured output images, and plot the relation between the experimental captured intensity and that of the original image. For both the *S*-shape and thresholding curve modulation, the experimental sampled points gather towards a certain area near the theoretical input-output transfer curve (lower panels of Fig. R4(a, b)). The root-mean square error (RMSE) defined as $RMSE = \sqrt{\frac{\sum_{i=1}^n (I_{experiment,i} - I_{theory,i})^2}{n}}$ (where $I_{experiment,i}$ and $I_{theory,i}$ are the experimental measured and the corresponding theoretic value of the output intensity, n is the total number sampled points) is used to quantify the discrepancy between the experimental data and the theoretical value. The RMSEs for the experimental data of the *S*-shape and thresholding curve are 0.15040, 0.14995, respectively, showing relative accuracy of the reconstructed images.

Fig. R4. Linearly gradient grayscale images (upper panels) and their input-output relations (lower panels) modulated by (a) the *S*-shape and (b) the thresholding curves.

We have added those experimental data analysis of the encoding-decoding procedure to quantify the accuracy in our reconstruction. Please refer to lines 7-9, 2nd paragraph, page 5 in the revised manuscript and Figure S14 in the revised supplementary material.

Response to Reviewer #3

GENERAL COMMENTS

This is nice fundamental work that generalizes Malus' law to project from one arbitrary elliptical polarization to another. An elegant formulation has been derived to mathematically express this generalized Malus' law, as shown in equations (1) and (2) in the main text. In this formulation, it was revealed that the local azimuthal parameter ψ_l can be used as a degree of freedom for polarization encoding following an arbitrary trajectory on the Poincare sphere. This theory has then been experimentally verified using a dedicated metasurface to showcase several analogue image processing experiments at lightwave frequencies.

In my opinion this work is excellent and of fundamental nature.

Moreover, the supporting experimental results are convincing. If I have some suggestion is to try and explain better the experimental process of encoding/decoding. I had hard time trying to understand this, despite the details provided in the Supplementary material (e.g. Figures S9, S10). How is the pixel-by-pixel encoding process implemented in practice? I suggest summarizing what is shown in S9,S10 with a short but descriptive text in the main body of the paper where the reference is made to Figures S9, S10. Perhaps also a figure in the Supplementary, like S11, which actually explains how the object is involved would be useful.

Response: We thank the reviewer for endorsing that our work is excellent and of fundamental nature and providing those important suggestions. To make it clear the pixel-by-pixel encoding and decoding process implemented in practice, we revised the Figures S9 and S10 to Figures S9-S12 (Figures R5-R8) to clearly illustrate the flowchart of the encoding/decoding process. As shown in Fig. R5(a), the intensity of each pixel of the image is used to determine the local ellipticity parameter χ_l of the polarization state, and the other local parameter ψ_l (azimuthal) is then determined as a function of χ_l , where the function could be engineered as a modulation trajectory between the local north and south poles of the Poincare sphere (lower-left panel Fig. R5(a)). After full local polarization parameters (ψ_l, χ_l) are both determined, they are transformed to global

polarization parameters (ψ, χ) by the analytic formulas,

$$\psi = \frac{1}{2} \arctan \left[\frac{\cos 2\chi_l \cos 2\psi_l \sin 2\chi_0 \sin 2\psi_0 + \cos 2\chi_l \sin 2\psi_l \cos 2\psi_0 + \sin 2\chi_l \cos 2\chi_0 \sin 2\psi_0}{\cos 2\chi_l \cos 2\psi_l \sin 2\chi_0 \cos 2\psi_0 - \cos 2\chi_l \sin 2\psi_l \sin 2\psi_0 + \sin 2\chi_l \cos 2\chi_0 \cos 2\psi_0} \right], \quad (16a)$$

$$\chi = \frac{1}{2} \arcsin[-\cos 2\chi_l \cos 2\psi_l \cos 2\chi_0 + \sin 2\chi_l \sin 2\chi_0]. \quad (16b)$$

Which is Eqs. (S20) in the supplementary materials, derived by the affine transformation between local/global parameters.

Fig. R5. The flowchart of the polarization information encoding with an analytic modulation trajectory. (a) The modulated image is mapped on the local ellipticity parameter $\chi_l(x, y)$ based on the local form of GML, while the local azimuth parameter is encoded by an analytic function $\psi_l(x, y) = f(\chi_l(x, y))$. (b) Then, the local PS parameters (ψ_l, χ_l) are transformed to global PS parameters (ψ, χ) with the information of polarization basis parameters (ψ_0, χ_0) . (c) After that, the global parameters (ψ, χ) are mapped to the meta-atom parameters for experimental realization.

Fig. R6. The flowchart of the decoding process with an analytic modulation trajectory. (a) In the decoder, illuminating the metasurface with circularly polarized incident light, a spatially varying profile of polarization states appears in the transmitted light beam. (b) Placing a PS universal analyzer with allowed polarization state (ψ_A, χ_A) on the transmitted light beam, the decoded intensity can be obtained by the global form of GML (c). Varying the analyzer state on the PS, the decoded intensity information will be different based on the modulated PS trajectories.

The global polarization parameters $(\psi(x,y), \chi(x,y))$ define the real polarization distribution as shown in Fig. R5(b) to be recorded by the physical medium. To encode the spatially-varying polarization distribution pixel by pixel, we employ the all-dielectric metasurface with anisotropic meta-atoms typically as rectangular nanopillars. The length D_x , width D_y and orientation angle θ are tailored to modulate the phase retardation $\Delta \varphi$ and angle θ of the effective local retarder that generate the predefined polarizations state, which obeys the following relation,

$$\theta = \psi + \frac{\pi}{4}, \Delta \varphi = \frac{\pi}{4} - \chi \Rightarrow (D_x, D_y). \quad (17)$$

In this way, both the input image and the modulation trajectory on the Poincaré sphere are encoded by a single layered metasurface.

In the decoding process, as shown in Fig. R6, the metasurface with varying nanopillar length, width, and orientation angle is placed on a PS universal polarizer,

which can be typically constructed by a quarter-waveplate (QWP) and a linear polarizer (LP), the allowed transmission polarization state (ψ_A, χ_A) can be set by rotating the QWP and LP with orientation angle, $\theta_{\lambda/4} = \psi_0$, $\theta_p = \psi_0 - \chi_0$, respectively [Fig. R6 (b)]. After the full Poincare sphere polarizer, the output intensity profile recorded by the CCD can be obtained by the global form of the GML,

$$I_{decode} = \frac{1}{2} [\cos 2\chi_A \cos 2\chi \cos(2\psi - 2\psi_A) + \sin 2\chi_A \sin 2\chi + 1], \quad (18)$$

where ψ, χ are polarization parameters for the input light, and ψ_A, χ_A are parameters of the analyzer allowed polarization state. By varying the analyzer parameters (ψ_A, χ_A) through combination of different rotating angles of QWP and LP $(\theta_{\lambda/4}, \theta_p)$, different input-output transfer relation appears, resulting different grayscale transformed output images.

For the double information channel encryption, the intensities of both the two images I_A, I_B are modulated by two sets of local ellipticity parameters χ_l and χ_l' . And then, those parameters (χ_l, χ_l') are transferred to global polarization parameters (ψ, χ) according to the following Poincare sphere equations,

$$\sin 2\chi_l = \cos 2\chi_0 \cos 2\psi_0 \cos 2\chi \cos 2\psi + \cos 2\chi_0 \sin 2\psi_0 \cos 2\chi \sin 2\psi + \sin 2\chi_0 \sin 2\chi, \quad (19a)$$

$$\sin 2\chi_l' = \cos 2\chi_0' \cos 2\psi_0' \cos 2\chi \cos 2\psi + \cos 2\chi_0' \sin 2\psi_0' \cos 2\chi \sin 2\psi + \sin 2\chi_0' \sin 2\chi. \quad (19b)$$

and then, the global polarization distribution is modulated by the all-dielectric anisotropic metasurface in the same way as the previous encoding process. The main decoding process of the double information channel encryption is similar with that of the previous single information channel approach, except that, the decoded images in the CCD reveals alternative images of the two encoded ones, and decoding locations of independent information channels are not necessarily orthogonal, and could be engineered at will on the Poincare sphere.

Fig. R7 The flowchart of the polarization information encoding for double information channel encryption. (a) In the encoder, two sets of polarization images ‘A’ and ‘B’ are mapped on two local PS systems with ellipticity parameter $\chi_l(x, y)$ and $\chi_l'(x, y)$, respectively. (b) Then, the two sets of local ellipticity $\chi_l(x, y)$ and $\chi_l'(x, y)$ are transformed to one sets of PS parameters (ψ, χ) with the information of polarization basis parameters (ψ_0, χ_0) and (ψ_0', χ_0') (c). After that, the global parameters (ψ, χ) are mapped to the meta-atom parameters for experimental realization.

Fig. R8. The flowchart of the decoding process for double information channel encryption. (a) In the decoder, illuminating the metasurface with circularly polarized incident light, a spatially varying profile of polarization states appears in the transmitted light beam. (b) Placing a PS universal analyzer with allowed polarization state (ψ_A, χ_A)

on the transmitted light beam, the decoded intensity can be obtained by the global form of GML (c). Varying the analyzer state along a circular path on the PS, the decoded images emerge as original, grayscale-reversed and mixed versions of ‘*A*’ and ‘*B*’ images at particular analyzer locations on the PS.

We also summarized what is shown in Figs. S9-S12 of the supplementary material, and a short but descriptive text in the main body of the paper, please refer to lines 6-17, 2nd paragraph, and page 4 in the revised manuscript.

Reviewer #1 (Remarks to the Author):

With regard to my first point previously, I appreciate that the authors have made an attempt to de-emphasize their contribution to this "generalized" Malus' Law. I also see that they have expanded their analytical relations to include the case of partial polarization. However, I still maintain that this is not a specific contribution of the present work - Malus' Law is already general. Malus' Law is simply a colloquial name for the projective nature of polarization optics and applies equally well to linear polarization states as it does to partially polarized, elliptical polarization states passing through partially diattenuating, elliptical polarization analyzers. This case is conventionally described in terms of a simple inner product of the four-element Stokes vector of a given polarization state with the four-element Stokes vector describing the analysis state of the polarizer.

What the authors have done in this work is to write the polarization state and the analyzer in terms of Jones vectors, explicitly expanding the projected intensity in terms of the angular parameters polarization state and analyzer. This is fine, but I disagree with the authors' claim that this is a significantly novel contribution of this work, nor do I agree that it aids intuition or design. The expressions obtained by the authors are a complex collection of trigonometric factors. Historically, this exact problem (polarization passing through a polarization analyzer) is what motivated the development of the Stokes vector/Poincare sphere formalism to begin with. Since it is formulated in power units (rather than the Jones vector which is formulated in terms of fields), the mathematical handling of projected intensity is much simpler, and one would generally prefer to use that formalism. (As the authors do show here, it is very clear that one could stick with the Jones calculus, but it becomes unwieldy and is generally not preferred).

As for my second point in the original review, I thank the authors for their reply. I appreciated the point made by Fig. R1 of the rebuttal letter even upon first reading the paper. The authors emphasize that their approach is useful for polarization "encoding" and provides the most general form thereof. However, I am not sure exactly which application requires this sort of encoding. I do not see a straightforward practical application here. In their rebuttal, the authors say that the "proposed Poincare sphere trajectory encoding approach is a general polarization information encoding methodology that could perform arbitrary nonlinear transformations of encoded polarization image in the metasurface". This statement is true, but from a technological or application standpoint, I am not sure why it is useful. The nonlinearity is just a reproduction of nonlinearity implemented digitally, during the design of the metasurface. I agree with the authors that this work represents a generalization of work performed under the heading of "Malus metasurfaces" in recent years, but I have the same concern with that body of work in general.

For these reasons, I still do not find the work to be of sufficient novelty for publication in Nature Communications.

Reviewer #2 (Remarks to the Author):

The authors have addressed all of comments and concerns.

Reviewer #3 (Remarks to the Author):

I thank the authors for providing more details on how the pixel-by-pixel encoding/decoding process is implemented in practice.

Re. NCOMMS-23-47496A

A detailed point-by-point response to all the reviewers' questions is provided in the following. The original remarks of the reviewers are reproduced in **black**; our responses are provided in **blue**.

Response to Reviewer #1

GENERAL COMMENTS

Comment 1) With regard to my first point previously, I appreciate that the authors have made an attempt to de-emphasize their contribution to this "generalized" Malus' Law. I also see that they have expanded their analytical relations to include the case of partial polarization. However, I still maintain that this is not a specific contribution of the present work - Malus' Law is already general. Malus' Law is simply a colloquial name for the projective nature of polarization optics and applies equally well to linear polarization states as it does to partially polarized, elliptical polarization states passing through partially diattenuating, elliptical polarization analyzers. This case is conventionally described in terms of a simple inner product of the four-element Stokes vector of a given polarization state with the four-element Stokes vector describing the analysis state of the polarizer.

What the authors have done in this work is to write the polarization state and the analyzer in terms of Jones vectors, explicitly expanding the projected intensity in terms of the angular parameters polarization state and analyzer. This is fine, but I disagree with the authors' claim that this is a significantly novel contribution of this work, nor do I agree that it aids intuition or design. The expressions obtained by the authors are a complex collection of trigonometric factors. Historically, this exact problem (polarization passing through a polarization analyzer) is what motivated the development of the Stokes vector/Poincare sphere formalism to begin with. Since it is formulated in power units (rather than the Jones vector which is formulated in terms of fields), the mathematical handling of projected intensity is much simpler, and one

would generally prefer to use that formalism. (As the authors do show here, it is very clear that one could stick with the Jones calculus, but it becomes unwieldy and is generally not preferred).

Response: We thank the reviewer for the careful reading of our work, and endorsing of our effort in clarifying and expanding the analytical form in terms of Jones vectors to further include the partial polarization scenario. We agree that the polarization projection law itself may not be a specific contribution of the present work, and appreciate the reviewer's effort of providing an historical context on the development of the Stokes vector and Poincaré sphere formalisms. We believe that the use of the polarization projection law for general polarizations and the realization of metasurfaces that can leverage this projection law is the main contribution of our work, and it is a nontrivial advance. We also agree that, the Stokes vector formalism with all components as power unit provide significant convenience to mathematically handle the intensity projection of arbitrary polarizations. The elements of the Stokes vector are directly related to the experimentally measured ones, providing convenience for experimental polarization measurements. From the perspective of metasurface design, the meta-atom parameters, including its length, width and orientation angle are closely related to the Jones vector polarization parameters, rather than the Stokes parameters. As shown in Supplementary Note 3 and Eqs. (S46)-(S50) and in the Supplementary Material, the modulation of light's properties by anisotropic meta-atoms are formulated in terms of complex-valued Jones matrix with both amplitude and phase responses. Therefore, we believe that, for general polarization measurements, the Stokes vector formalism is preferable, since in terms of the polarization encoding metasurface design the Jones matrix treatment of the polarization projection behavior is more convenient. In addition, the angular parameters of the polarization state provide flexibility in building a local Poincaré sphere system, in which the hidden dimension in the projection relation can emerge as a new degree of freedom, adding new elements to the design of polarization encoding metasurfaces. The explicit expressions in Jones calculus for the polarization projection relation in both local and global forms are directly used in polarization encoding and decoding processes, as shown in Fig. S9 and Fig. S10 in the revised

supplementary materials, which we believe are more straightforward and simpler than the Stokes vector treatment. To address this point, we have added related description in the last 7 lines, Paragraph 3, Page 3 of the revised manuscript.

Comment 2) As for my second point in the original review, I thank the authors for their reply. I appreciated the point made by Fig. R1 of the rebuttal letter even upon first reading the paper. The authors emphasize that their approach is useful for polarization "encoding" and provides the most general form thereof. However, I am not sure exactly which application requires this sort of encoding. I do not see a straightforward practical application here. In their rebuttal, the authors say that the "proposed Poincaré sphere trajectory encoding approach is a general polarization information encoding methodology that could perform arbitrary nonlinear transformations of encoded polarization image in the metasurface". This statement is true, but from a technological or application standpoint, I am not sure why it is useful. The nonlinearity is just a reproduction of nonlinearity implemented digitally, during the design of the metasurface. I agree with the authors that this work represents a generalization of work performed under the heading of "Malus metasurfaces" in recent years, but I have the same concern with that body of work in general.

Response: We thank the reviewer for highlighting that the proposed Poincaré sphere trajectory encoding approach is a general polarization information encoding methodology that can perform arbitrary nonlinear transformations of encoded polarization images using metasurfaces, and that this work represents a generalization of work performed under the heading of "Malus metasurfaces" in recent years. As the present work mainly introduces the general encoding approach, it does not directly aim at a specific application. Specific applications based on such general approaches will be considered in our following work. However, these polarization information-encoding approaches are typically used in optical encryption and anti-counterfeiting applications. The polarization encoded image manifests hidden information behind a uniform intensity profile, which naturally provides information hiding and encryption

functionalities. Multiple different polarization encoded images by metasurfaces are required for the cipher images and the key for both the encryption and the decryption process. A recent specific application-oriented Malus metasurface paper [Sci. Adv. 2021; 7: eabg0363, (Ref.33 in the main text of the manuscript)] shows that, by combining such polarization encoded meta-image with other computational imaging algorithms such as single-pixel-imaging techniques, the pattern-fixed Malus metasurface can even become a reusable encryption device for practical applications. Because the polarization-encoded meta-image plays the role of an encryption matrix, and different operations on the matrix yield different steganographic information channels, a single metasurface with fixed meta-image can be used in multiple round encryptions. Although our present work does not directly demonstrate the computational imaging assisted encryption and anticounterfeiting application, our proposed general polarization encoding approach can be straightforwardly employed in those applications, and significantly increase the information amount delivered by a single metasurface and hence enhance the security level, as the polarization encoding diversity is largely promoted by a dimension raising from the 1D to 2D projection. Therefore, our proposed general polarization encoding approach indeed can find implications as a reusable encryption and/or anti-counterfeiting device when it is combined with advanced computational imaging techniques, promising for practical applications. We have clarified the practical applications of our proposed approach on lines 9-13, paragraph 2, Page 6 in the revised manuscript.

We believe that, with these clarifications, the paper meets the standards for publication in Nature Communications, and we hope the reviewer may agree with us and the other reviewers.

Response to Reviewer #2

GENERAL COMMENTS

The authors have addressed all of comments and concerns.

Response: We thank the reviewer for endorsing our paper.

Response to Reviewer #3

GENERAL COMMENTS

I thank the authors for providing more details on how the pixel-by-pixel encoding/decoding process is implemented in practice.

Response: We thank the reviewer for endorsing our paper.